# MASSIVELY PARALLEL HYPERPARAMETER TUNING

## ABSTRACT

Modern learning models are characterized by large hyperparameter spaces. In order to adequately explore these large spaces, we must evaluate a large number of configurations, typically orders of magnitude more configurations than available parallel workers. Given the growing costs of model training, we would ideally like to perform this search in roughly the same wall-clock time needed to train a single model. In this work, we tackle this challenge by introducing ASHA, a simple and robust hyperparameter tuning algorithm with solid theoretical underpinnings that exploits parallelism and aggressive early-stopping. Our extensive empirical results show that ASHA outperforms state-of-the-art hyperparameter tuning methods; scales linearly with the number of workers in distributed settings; converges to a high quality configuration in half the time taken by Vizier (Google's internal hyperparameter tuning service) in an experiment with 500 workers; and beats the published result for a near state-of-the-art LSTM architecture in under $2\times$ the time to train a single model.

## 1 INTRODUCTION

Although machine learning models have recently achieved dramatic successes in a variety of practical applications, these models are highly sensitive to internal parameters, i.e., *hyperparameters*. In these modern learning regimes, three trends motivate a new approach to hyperparameter tuning:

1. **High-dimensional hyperparameter spaces**. Machine learning models are becoming increasingly complex, as evidenced by modern neural networks with dozens of hyperparameters. For such complex models with hyperparameters that interact in unknown ways, a practitioner is forced to evaluate potentially thousands of different hyperparameter settings.
2. **Increasing training times**. As datasets grow larger and models become more complex, training a model has become dramatically more expensive, often taking days or weeks on specialized high-performance hardware. This trend is particularly onerous in the context of hyperparameter tuning, as a new model must be trained to evaluate each candidate hyperparameter configuration.
3. **Rise of parallel computing**. The combination of a growing number of hyperparameters and longer training time per model precludes evaluating configurations sequentially; we simply cannot wait months or years to find a suitable hyperparameter setting. Leveraging parallel and distributed computational resources presents a solution to the increasingly challenging problem of hyperparameter optimization.

Our goal is to design a hyperparameter tuning algorithm that can effectively leverage parallel resources. This goal seems trivial with standard methods like random search, where we could train different configurations in an embarrassingly parallel fashion. However, for high-dimensional search spaces, the number of candidate configurations required to find a good configuration often dwarfs the number of available parallel resources, and when possible, our goal is to:

*Evaluate orders of magnitude more hyperparameter configurations than available parallel workers in a small multiple of the wall-clock time needed to train a single model.*

We denote this setting as the *large-scale regime* for hyperparameter optimization. In this work we study this regime, and our contributions are as follows:

- We introduce the **A**synchronous **S**uccessive **H**alving **A**lgorithm (ASHA), a practical hyperparameter tuning method for the large-scale regime that exploits parallelism and aggressive early-stopping.

Our algorithm is inspired by the Successive Halving algorithm (SHA) (Karnin et al., 2013; Jamieson & Talwalkar, 2015), a theoretically principled early-stopping method that allocates more resources to promising configurations. As a side benefit of adapting SHA for the parallel setting, we also improve upon the synchronous version of SHA, as described in Section 3.3.

- In the sequential setting, we compare SHA with BOHB (Falkner et al., 2018), Fabolas (Klein et al., 2017), and Population Based Tuning (PBT) (Jaderberg et al., 2017), state-of-the-art methods that exploit partial training. Our results show that SHA slightly outperforms these methods, which when coupled with SHA's simplicity and theoretical grounding, motivate the use of SHA in parallel settings. We also verify that SHA and ASHA achieve similar results.

- In a distributed setting with 25 workers, we evaluate ASHA's suitability for the large-scale regime. We focus on tuning two CNNs on CIFAR-10, and observe that ASHA finds a good configuration in approximately the time it takes to train a single model. We also show that ASHA scales linearly with the number of workers, and that it again slightly exceeds the performance of PBT and BOHB, a method combining Bayesian optimization with SHA.

- In a distributed setting with 500 workers, we compare ASHA to the default tuning algorithm used by Vizier (Golovin et al., 2017), Google's internal hyperparameter tuning service. We show that ASHA outperforms Vizier when tuning an LSTM model on the Penn Treebank dataset (PTB).

- Finally, we compare ASHA and PBT on a task using 16 GPUs to tune a near state-of-the-art language model (Merity et al., 2018), with a published test perplexity of 57.3 on PTB. ASHA outperforms PBT and is able to find a configuration with a test perplexity of 56.3 in less than $2\times$ the time it takes to train a single model.

## 2 RELATED WORK

**Sequential methods**: There is a large body of work in this area, with existing methods focusing on adaptively evaluating configurations, adaptively selecting configurations, or taking a hybrid approach that combines both strategies. SHA (Jamieson & Talwalkar, 2015) and Hyperband (Li et al., 2018) are empirically effective adaptive evaluation techniques that employ principled 'early-stopping.' SHA serves as the inner loop for Hyperband, with Hyperband automating the choice of the early-stopping rate by running different variants of SHA. While the appropriate choice of early stopping rate is problem dependent, Li et al. (2018)'s empirical results show that aggressive early-stopping works well for a wide variety of tuning tasks. Hence, we focus on adapting SHA to the parallel setting in Section 3, though we also evaluate the corresponding asynchronous Hyperband method.

Fabolas (Klein et al., 2017), a hybrid Bayesian optimization method, has demonstrated state-of-the-art performance on several tasks in comparison to Hyperband and other leading methods. While Fabolas, along with most other Bayesian optimization approaches, can be parallelized using a constant liar (CL) type heuristic (Ginsbourger et al., 2010; González et al., 2016), the parallel version will underperform the sequential version, since the latter uses a more accurate posterior to propose new points. Hence, we compare to Fabolas in the sequential setting in Section 4.1, and demonstrate that under an appropriate experimental setup, SHA and Hyperband in fact slightly outperform Fabolas.

More recently in parallel work, Falkner et al. (2018) introduced BOHB, a hybrid method combining Bayesian optimization with Hyperband. They also propose a parallelization scheme for SHA that retains synchronized eliminations of underperforming configurations. We discuss the drawbacks of this parallelization scheme in Section 3 and demonstrate that ASHA outperforms this version of parallel SHA as well as BOHB in Section 4.2.

**Parallel methods**: Jaderberg et al. (2017) developed PBT, a state-of-the-art hybrid evolutionary approach that exploits partial training to iteratively increase the fitness of a population of models. Additionally, Golovin et al. (2017) introduced Vizier, a black-box optimization service with support for multiple tuning methods and early-stopping options. For succinctness, we will refer to Vizier's default algorithm as "Vizier" with the understanding that it is simply one of methods available on the Vizier platform. We compare to PBT and Vizier in Section 4.2 and Section 4.3, respectively. We further note that PBT is more comparable to SHA than to Hyperband since both PBT and SHA require the user to set the early-stopping rate via internal hyperparameters.

**Neural Architecture Search (NAS) methods**: Another class of related methods are tailored for search spaces designed to tune the architectures of neural networks. While earlier methods were

computationally expensive (Zoph & Le, 2017; Real et al., 2017), more recent methods exploit reuse to reduce the costs (Pham et al., 2018; Bender et al., 2018). However, these methods suffer from reproducibility issues (Pham et al., 2018; Pham & Guan, 2018), and recent work has demonstrated that appropriately tuning more traditional architectures can match or outperform the results from architecture search (Melis et al., 2018). Relatedly, ASHA's performance in Section 4.3.1 matches that of leading NAS methods on the PTB benchmark (Pham et al., 2018; Pham & Guan, 2018).

# 3 ASHA ALGORITHM

We start with an overview of SHA (Karnin et al., 2013; Jamieson & Talwalkar, 2015) and motivate the need to adapt it to the parallel setting. Then we present ASHA and discuss how it addresses issues with synchronous SHA and improves upon the original algorithm.

## 3.1 SUCCESSIVE HALVING (SHA)

---

**Algorithm 1:** Successive Halving Algorithm.

1 **Input:** number of configurations $n$, minimum resource $r$, maximum resource $R$, reduction factor $\eta$, minimum early-stopping rate $s$
2 $s_{\max} = \lfloor \log_\eta(R/r) \rfloor$
3 `assert` $n \geq \eta^{s_{\max}-s}$ so that at least one configuration will be allocated $R$.
4 $T = \texttt{get\_hyperparameter\_configuration}(n)$
5 **for** $i \in \{0, \ldots, s_{\max} - s\}$ **do**
6     `// All configurations trained for a given i constitute a "rung."`
7     $n_i = \lfloor n\eta^{-i} \rfloor$
8     $r_i = r\eta^{i+s}$
9     $L = \texttt{run\_then\_return\_val\_loss}(\theta, r_i) : \theta \in T$
10     $T = \texttt{top\_k}(T, L, n_i/\eta)$
11 **end**
12 **return** *best configuration in $T$*

---

The idea behind SHA (Algorithm 1) is simple: allocate a small budget to each configuration, evaluate all configurations and keep the top $1/\eta$, increase the budget per configuration by a factor of $\eta$, and repeat until the maximum per-configuration budget of $R$ is reached (lines 5–11). The resource allocated by SHA can be iterations of stochastic gradient descent, number of training examples, number of random features, etc.

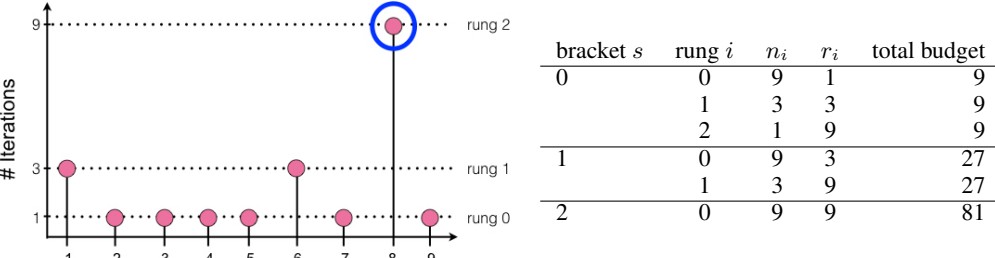

| bracket $s$ | rung $i$ | $n_i$ | $r_i$ | total budget |
|---|---|---|---|---|
| 0 | 0 | 9 | 1 | 9 |
| | 1 | 3 | 3 | 9 |
| | 2 | 1 | 9 | 9 |
| 1 | 0 | 9 | 3 | 27 |
| | 1 | 3 | 9 | 27 |
| 2 | 0 | 9 | 9 | 81 |

Figure 1: **Promotion scheme for SHA** with $n = 9, r = 1, R = 9$, and $\eta = 3$. (Left) Visual depiction of the promotion scheme for bracket $s = 0$. (Right) Promotion scheme for different brackets $s$. $s$ increases the starting budget per configuration $r_0 \leq R$ by a factor of $\eta$ per increment of $s$. Hence, it takes more resources to explore the same number of configurations for higher $s$. Note that for a given $s$, the same budget is allocated to each rung but is split between fewer configurations in higher rungs.

SHA requires the number of configurations $n$, a minimum resource $r$, a maximum resource $R$, a reduction factor $\eta \geq 2$, and a minimum early-stopping rate $s$. Additionally, the

`get_hyperparameter_configuration`$(n)$ subroutine returns $n$ configurations sampled randomly from a given search space; and the `run_then_return_val_loss`$(\theta, r)$ subroutine returns the validation loss after training the model with the hyperparameter setting $\theta$ and for $r$ resources. For a given early-stopping rate $s$, a minimum resource of $r_0 = r\eta^s$ will be allocated to each configuration. Hence, lower $s$ corresponds to more aggressive early-stopping, with $s = 0$ prescribing a minimum resource of $r$. We will refer to SHA with different values of $s$ as *brackets* and, within a bracket, we will refer to each round of promotion as a *rung* with the base rung numbered 0 and increasing. Figure 1(left) shows the rungs for bracket 0 for an example setting with $n = 9, r = 1, R = 9$, and $\eta = 3$, while Figure 1(right) shows how resource allocations change for different brackets.

The naive way of parallelizing SHA is to distribute the training of the $n/\eta^k$ surviving configurations on each rung $k$ as is done by Falkner et al. (2018) and add brackets when there are no jobs available in existing brackets. We will refer to this method as "synchronous" SHA. The efficacy of this strategy is severely hampered by two issues: (1) SHA's synchronous nature is sensitive to stragglers and dropped jobs as every configuration within a rung must complete before proceeding to the next rung, and (2) the estimate of the top $1/\eta$ configurations for a given early-stopping rate does not improve as more brackets are run since promotions are performed independently for each bracket. We demonstrate in Appendix A.1 that ASHA is more robust than synchronous SHA to stragglers and dropped jobs using simulated workloads. Nonetheless, we compare synchronous SHA in Section 4.3 and show that ASHA performs better.

We could also consider the embarrassingly parallel approach of running multiple instances of SHA, one on each worker. However, this strategy is not well suited for the large-scale regime, where we would like results in little more than the time to train one configuration. To see this, assume that training time for a configuration scales linearly with the allocated resource and $time(R)$ represents the time required to train a configuration for the maximum resource $R$. In general, for a given bracket $s$, the minimum time to return a configuration trained to completion is $(\log_\eta(R/r) - s + 1) \times time(R)$, where $\log_\eta(R/r) - s + 1$ counts the number of rungs. For example, consider Bracket 0 in the toy example in Figure 1. The time needed to return a fully trained configuration is $3 \times time(R)$, since there are three rungs and each rung is allocated $R$ resource. Moreover, using this parallelization strategy on the task considered in Section 4.3.1 would have required 5 days to return an answer, compared to the 1 day needed for ASHA.

---

**Algorithm 2:** Asynchronous Successive Halving Algorithm.

1  **Input:** minimum resource $r$, maximum resource $R$, reduction factor $\eta$, minimum early-stopping rate $s$
2  **Algorithm** `ASHA()`
3    **repeat**
4      **for** each free worker **do**
5        $(\theta, k) = $ `get_job()`
6        `run_then_return_val_loss`$(\theta, r\eta^{s+k})$
7      **end**
8      **for** completed job $(\theta, k)$ with loss $l$ **do**
9        Update configuration $\theta$ in rung $k$ with loss $l$.
10      **end**
11  **Procedure** `get_job()`
12    // Check to see if there is a promotable config.
13    **for** $k = \lfloor \log_\eta(R/r) \rfloor - s, \ldots, 1, 0$ **do**
14      candidates $= $ `top_k`(rung $k$, |rung $k$|/$\eta$)
15      promotable $= \{t$ for $t \in$ candidates if $t$ not already promoted$\}$
16      **if** |promotable| $> 0$ **then**
17        **return** promotable$[0], k + 1$
18      **end**
19    **end**
20    Draw random configuration $\theta$. // If not, grow bottom rung.
21    **return** $\theta, 0$

---

## 3.2 ASYNCHRONOUS SHA (ASHA)

We now introduce ASHA as an effective technique to parallelize SHA, leveraging asynchrony to mitigate stragglers and maximize parallelism. Intuitively, ASHA promotes configurations to the next rung whenever possible instead of waiting for a rung to complete before proceeding to the next rung. Additionally, if no promotions are possible, ASHA simply adds a configuration to the base rung, so that more configurations can be promoted to the upper rungs. ASHA is formally defined in Algorithm 2. Given its asynchronous nature it does not require the user to pre-specify the number of configurations to evaluate, but it otherwise requires the same inputs as SHA. Note that the `run_then_return_val_loss` subroutine in `ASHA` is asynchronous and the code execution continues after the job is passed to the worker. ASHA's promotion scheme is laid out in the `get_job` subroutine. We compare the promotion schemes of SHA and ASHA in Figure 2.

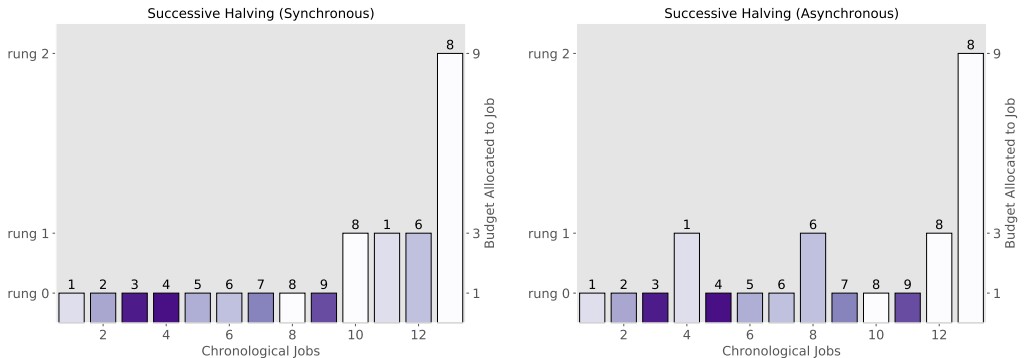

Figure 2: **Comparison of promotion schemes** for SHA and ASHA. We show the promotion schemes for Bracket 0 from Figure 1 (i.e. $r = 1, R = 9, \eta = 3$, and $s = 0$). Each bar represents a job along with the associated training budgets. The jobs submitted by SHA and ASHA are shown in chronological order going from left to right. The bars are labeled with the configuration number and the color gradient corresponds to performance, with lighter bars having lower error; as in Figure 1(left), configurations 1, 6, and 8 are promoted to rung 1 and configuration 8 is promoted to rung 2. SHA must complete all jobs in a rung before moving the next rung. In contrast, ASHA strives to maintain the simple rule that each rung should always have about $1/\eta = 1/3$ of the configurations as the rung below it, while new configurations are added to the bottom rung as needed.

ASHA is well-suited for the large-scale regime, where wall-clock time is constrained to a small multiple of the time needed to train a single model. For ease of comparison with SHA, assume training time scales linearly with the resource. Consider the example of Bracket 0 shown in Figure 1, and assume we can run ASHA with 9 machines. Then ASHA returns a fully trained configuration in $^{13}/_9 \times time(R)$, since 9 machines are sufficient to promote configurations to the next rung in the same time it takes to train a single configuration in the rung. Hence, the training time for a configuration in rung 0 is $1/9 \times time(R)$, for rung 1 it is $1/3 \times time(R)$, and for rung 2 it is $time(R)$. In general, $\eta^{\log_\eta(R)-s}$ machines are needed to advance a configuration to the next rung in the same time it takes to train a single configuration in the rung, and it takes $\eta^{s+i-\log_\eta(R)} \times time(R)$ to train a configuration in rung $i$. Hence, ASHA can return a configuration trained to completion in time

$$\left( \sum_{i=s}^{\log_\eta(R)} \eta^{i-\log_\eta(R)} \right) \times time(R) \leq 2\,time(R).$$

Moreover, when training is iterative, ASHA can return an answer in $time(R)$, since incrementally trained configurations can be checkpointed and resumed.

Finally, since Hyperband simply runs multiple SHA brackets, we can asynchronously parallelize Hyperband by either running multiple brackets of ASHA or looping through brackets of ASHA sequentially as is done in the original Hyperband. We employ the latter looping scheme for asynchronous Hyperband in the next section.

### 3.3 Algorithm Discussion

ASHA is able to remove the bottleneck associated with synchronous promotions at the cost of a small number of incorrect promotions. By the law of large numbers, we expect to erroneously promote a vanishing fraction of configurations in each rung as the number of configurations grows. Intuitively, in the first rung with $n$ evaluated configurations, the number of mispromoted configurations is roughly $\sqrt{n}$, since the process resembles the convergence of an empirical cumulative distribution function to its expected value (c.f., the Dvoretzky-Kiefer-Wolfowitz inequality (Dvoretzky et al., 1956)). For later rungs, although the configurations are no longer i.i.d. since they were advanced based on the empirical CDF of the evaluation scores, we expect this dependence to be weak.

We further note that ASHA improves upon SHA in two ways. First, Li et al. (2018) discusses two SHA variants: finite horizon (bounded resource per configuration, i.e., $R$ is bounded) and infinite horizon (unbounded resources per configuration, i.e., $R$ is unbounded). ASHA consolidates these settings into one algorithm. In Algorithm 2, we do not promote configurations that have been trained for $R$, thereby restricting the number of rungs. However, this algorithm trivially generalizes to the infinite horizon; we can remove this restriction so that the maximum resource per configuration increases naturally as configurations are promoted to higher rungs. In contrast, SHA does not naturally extend to the infinite horizon setting, as it relies on the doubling trick and must rerun brackets with larger budgets to increase the maximum resource.

Additionally, SHA does not return an output until a single bracket completes. In the finite horizon this means that there is a constant interval of (# of rungs $\times time(R)$) between receiving outputs from SHA. In the infinite horizon this interval doubles between outputs. In contrast, ASHA grows the bracket incrementally instead of in fixed budget intervals. To further reduce latency, ASHA uses intermediate losses to determine the current best performing configuration, as opposed to only considering the final SHA outputs.

## 4 Empirical Evaluation

We first present results in the sequential setting to justify our choice of focusing on SHA and to compare SHA to ASHA. We next evaluate ASHA in parallel environments on three benchmark tasks.

### 4.1 Sequential Experiments

We benchmark Hyperband and SHA against PBT, BOHB (synchronous SHA with Bayesian optimization as introduced by Falkner et al. (2018)), and Fabolas, and examine the relative performance of SHA versus ASHA and Hyperband versus asynchronous Hyperband. As mentioned previously, asynchronous Hyperband loops through brackets of ASHA with different early-stopping rates, switching brackets when a budget corresponding to a hypothetical bracket of SHA would be depleted.

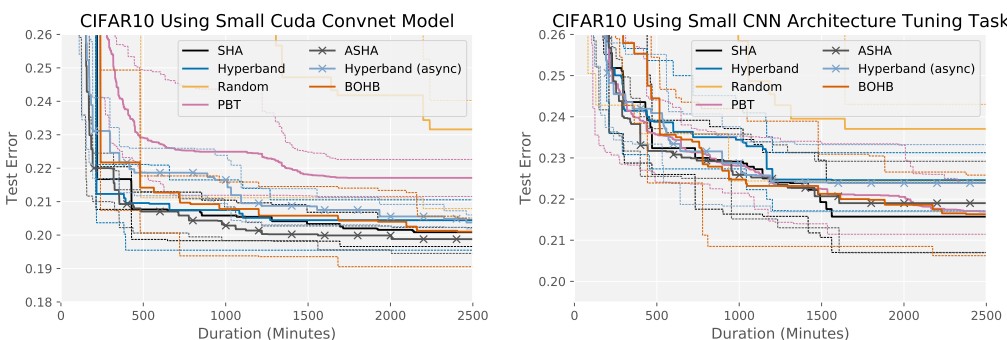

Figure 3: **Sequential experiments** (1 worker). Average across 10 trials is shown for each hyperparameter optimization method. Gridded lines represent top and bottom quartiles of trials.

We compare ASHA against PBT, BOHB, and synchronous SHA on two benchmarks for CIFAR-10: (1) tuning a convolutional neural network (CNN) with the cuda-convnet architecture and the same

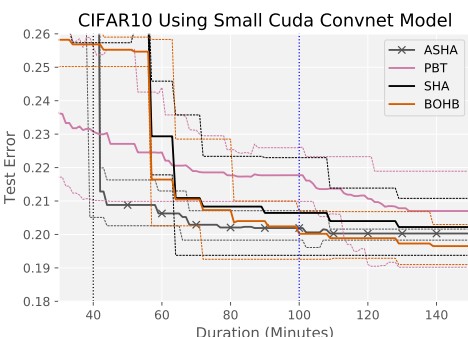 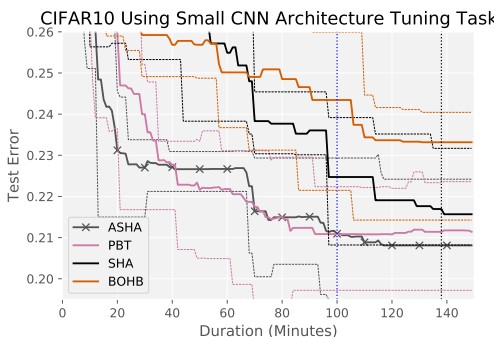

Figure 4: **Limited-scale distributed experiments** with 25 workers. For each searcher, the average test error across 5 trials is shown in each plot. The light dashed lines indicate the min/max ranges. The dotted black line represents the time needed to train the most expensive model in the search space for the maximum resource $R$. The dotted blue line represents the point at which 25 workers in parallel have performed as much work as a single machine in the sequential experiments (Figure 3).

search space as Li et al. (2017); and (2) tuning a CNN architecture with varying number of layers, batch size, and number of filters. The details for the search spaces considered for each benchmark and the settings we used for each search method can be found in Appendix A.3. Note that BOHB uses SHA to perform early-stopping and differs only in how configurations are sampled; while SHA uses random sampling, BOHB uses Bayesian optimization to adaptively sample new configurations. In the following experiments, we run BOHB using the same early-stopping rate as SHA and ASHA instead of looping through brackets with different early-stopping rates as is done by Hyperband.

The results on these two benchmarks are shown in Figure 3. On benchmark 1, Hyperband and all variants of SHA (i.e., SHA, ASHA, and BOHB) outperform PBT by $3\times$. On benchmark 2, while all methods comfortably beat random search, SHA, ASHA, BOHB and PBT performed similarly and slightly outperform Hyperband and asynchronous Hyperband. This last observation (i) corroborates the results in Li et al. (2017), which found that the brackets with the most aggressive early-stopping rates performed the best; and (ii) follows from the discussion in Section 2 noting that PBT is more similar in spirit to SHA than Hyperband, as PBT / SHA both require user-specified early-stopping rates (and are more aggressive in their early-stopping behavior in these experiments). We observe that SHA and ASHA are competitive with BOHB, despite the adaptive sampling scheme used by BOHB.

For both tasks, introducing asynchrony does not consequentially impact the performance of ASHA (relative to SHA) or asynchronous Hyperband (relative to Hyperband). This not surprising; as discussed in Section 3.3, we expect the number of ASHA mispromotions to be square root in the number of configurations.

Finally, due to the nuanced nature of the evaluation framework used by Klein et al. (2017) in comparing Fabolas with Hyperband, we present our results on 4 different benchmarks along with a detailed discussion of the experimental setup in Appendix A.2. In summary, our results show that Hyperband, specifically the first bracket of SHA, tends to outperform Fabolas while also exhibiting lower variance across experimental trials.

## 4.2 LIMITED-SCALE DISTRIBUTED EXPERIMENTS

We next compare ASHA to synchronous SHA, the parallelization scheme discussed in Section 3.1; BOHB; and PBT on the same two tasks from Section 4.1. For each experiment, we run each search method with 25 workers for 150 minutes. We use the same setups for ASHA and PBT as in the previous section. We run synchronous SHA and BOHB with default settings and the same $\eta$ and early-stopping rate as ASHA. Figure 4 shows the average test error across 5 trials for each search method. On both tasks, ASHA succeeded in the large-scale regime. For benchmark 1, ASHA evaluated over 1000 configurations in just over 40 minutes with 25 workers and found a good configuration (error rate below 0.21) in approximately the time needed to train a single model, whereas it took ASHA nearly 400 minutes to do so in the sequential setting (Figure 3). Notably, we only achieve a $10\times$

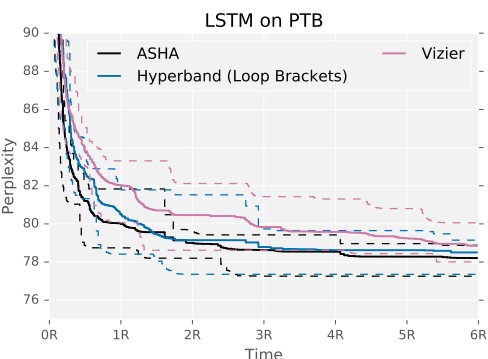

Figure 5: **Large-scale ASHA benchmark** that takes on the order of weeks to run with 500 workers. The x-axis is measured in units of average time to train a single configuration for $R$ resource. The average across 5 trials is shown, with dashed lines indicating min/max ranges.

speedup on 25 workers due to the relative simplicity of this task, i.e., it only required evaluating a few hundred configurations before identifying a good one in the sequential setting. In contrast, when considering the more difficult search space in benchmark 2, we observe linear speedups with ASHA, as the roughly 700 minutes in the sequential setting (Figure 3) needed to find a configuration with test error below $0.23$ is reduced to under 25 minutes in the distributed setting.

Compared to synchronous SHA and BOHB, ASHA finds a good configuration $1.5\times$ as fast on benchmark 1 while BOHB finds a slightly better final configuration. On benchmark 2, ASHA performs significantly better than synchronous SHA and BOHB due to the higher variance in training times between configurations (the average time required to train a configuration on the maximum resource $R$ is 30 minutes with a standard deviation of 27 minutes), which exacerbates the sensitivity of synchronous SHA to stragglers (see Appendix A.1). BOHB actually underperforms synchronous SHA on benchmark 2 due to its bias towards more computationally expensive configurations, reducing the number of configurations trained to completion within the given time frame.

We further note that ASHA outperforms PBT on benchmark 1; in fact the minimum and maximum range for ASHA across 5 trials does not overlap with the average for PBT. On benchmark 2, PBT slightly outperforms asynchronous Hyperband and performs comparably to ASHA. However, note that the ranges for the searchers share large overlap and the result is likely not significant. Overall, ASHA outperforms PBT, BOHB and SHA on these two tasks. This improved performance, coupled with the fact that it is a more principled and general approach than either BOHB or PBT (e.g., agnostic to resource type and robust to hyperparameters that change the size of the model), further motivates its use for the large-scale regime.

### 4.3 TUNING LARGE-SCALE LANGUAGE MODELS

We tune a one layer LSTM language model for next word prediction on the Penn Treebank dataset (Marcus et al., 1993). Our search space is constructed based off of the LSTMs considered in Zaremba et al. (2014), with the largest model in our search space matching their large LSTM (see Appendix A.5 for more details). Each tuner is given 500 workers and $6 \times time(R)$, i.e., $6\times$ the average time needed to train a single model. For ASHA, we set $\eta = 4$, $r = {}^{R}/64$, and $s = 0$; asynchronous Hyperband loops through brackets $s = 0, 1, 2, 3$. We compare to Vizier without the performance curve early-stopping rule (Golovin et al., 2017).[1]

The results in Figure 5 show that ASHA and asynchronous Hyperband found good configurations for this task in $1 \times time(R)$. Additionally, ASHA and asynchronous Hyperband are both about $3\times$ faster than Vizier at finding a configuration with test perplexity below 80, despite being much simpler and easier to implement. Furthermore, the best model found by ASHA achieved a test perplexity of

---

[1] At the time of running the experiment, it was brought to our attention by the team maintaining the Vizier service that the early-stopping code contained a bug. The bug negatively impacted the performance of Vizier with early-stopping; hence we omit the results here.

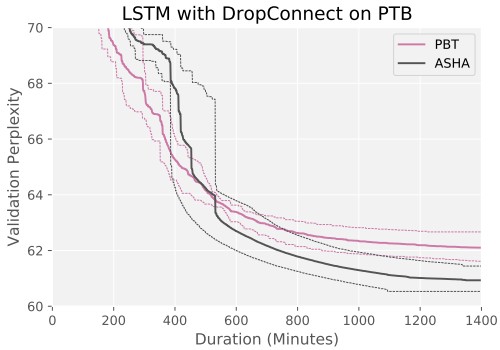

Figure 6: **Modern LSTM benchmark** with DropConnect (Merity et al., 2018) using 16 GPUs. The average across 5 trials is shown, with dashed lines indicating min/max ranges.

76.6, which is significantly better than 78.4 reported for the large LSTM in Zaremba et al. (2014). We also note that asynchronous Hyperband initially lags behind ASHA, but eventually catches up at around $1.5 \times time(R)$.

Notably, we observe that certain hyperparameter configurations in this benchmark induce perplexities that are orders of magnitude larger than the average case perplexity. Model-based methods that make assumptions on the data distribution, such as Vizier, can degrade in performance without further care to adjust this signal. We attempted to alleviate this by capping perplexity scores at 1000 but this still significantly hampered the performance of Vizier. We view robustness to these types of scenarios as an additional benefit of ASHA and Hyperband.

### 4.3.1 Tuning Modern LSTM Architectures

We next performed a follow up experiment on the Penn Treebank dataset focusing on models with improved perplexity. Our starting point was the work of Merity et al. (2018), which introduced a near state-of-the-art LSTM architecture with DropConnect, a new regularization scheme that can be easily combined with existing LSTM implementations in common deep learning frameworks. We constructed a search space around their configuration and ran ASHA and PBT, each with 16 GPUS on one `p2.16xlarge` instance on AWS. For ASHA, we used $\eta = 4$, $r = 1$ epoch, $R = 256$ epochs, and $s = 0$. For PBT, we use a population size to 20, a maximum resource of 256 epochs, and perform explore/exploit every 8 epochs using the same settings as the previous experiments.

Figure 6 shows that while PBT performs better initially, ASHA soon catches up and finds a better final configuration; in fact, the min/max ranges for ASHA and PBT do not overlap at the end. We then trained the best configuration found by ASHA for more epochs and reached validation and test perplexities of 60.2 and 58.1 respectively before fine-tuning and 58.7 and 56.3 after fine-tuning. For reference, Merity et al. (2018) reported validation and test perplexities respectively of 60.7 and 58.8 without fine-tuning and 60.0 and 57.3 with fine-tuning. This demonstrates the effectiveness of ASHA in the large-scale regime for modern hyperparameter optimization problems.

## 5 Conclusion

In this paper, we introduced ASHA and demonstrated its suitability for the large-scale regime of hyperparameter optimization. Directions for future work include combining ASHA with adaptive selection methods and incorporating meta-learning to inform early-stopping. These extensions overlap with those for synchronous SHA, and any progress made for the synchronous algorithm will likely also apply for ASHA.

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

# A   APPENDIX

As part of our supplementary material, we (1) demonstrate the impact of stragglers and dropped jobs on synchronous SHA and ASHA, (2) present the comparison to Fabolas in the sequential setting, and (3) provide additional details for the empirical results shown in Section 4.

## A.1   SIMULATED WORKLOADS TO EVALUATE THE IMPACT OF STRAGGLERS AND DROPPED JOBS

For our simulated workloads, we run SHA with $\eta = 4$, $r = 1$, $R = 256$, and $n = 256$ and ASHA with the same values and the maximum early-stopping rate $s = 0$. Note that BOHB is also susceptible to stragglers and dropped jobs since it uses synchronous SHA as its parallelization scheme but leverages Bayesian optimization to perform adaptive sampling. We assume that the expected training time for each job is the same as the allocated resource. We simulated stragglers by multiplying the expected train time by $(1 + |z|)$ where $z$ is drawn from a normal distribution with mean 0 and a specified standard deviation. We simulated dropped jobs by assuming that there is a given $p$ probability that a job will be dropped at each time unit, hence, for a job with a runtime of 256 units, the probability that it is not dropped is $(1 - (1 - p)^{256}$. For each combination of training standard deviation of drop probability, we simulate ASHA and synchronous SHA 25 times.

Figure 7 shows that ASHA trains many more configurations to completion when the standard deviation is high; we hypothesis that this is one reason ASHA performs significantly better than synchronous SHA and BOHB for the second benchmark in Section 4.2. Figure 8 shows that ASHA returns a configuration trained for the maximum resource $R$ much faster than synchronous SHA when there is high variability in training time (i.e., stragglers) and dropped jobs.

## A.2   COMPARISON WITH FABOLAS IN SEQUENTIAL SETTING

Klein et al. (2017) showed that Fabolas can be over an order of magnitude faster than existing Bayesian optimization methods. Additionally, the empirical studies presented in Klein et al. (2017) suggest that Fabolas is faster than Hyperband at finding a good configuration. We conducted our own experiments to compare Fabolas with Hyperband on the following tasks:

1. Tuning an SVM using the same search space as Klein et al. (2017).
2. Tuning a convolutional neural network (CNN) with the same search space as Li et al. (2017) on CIFAR-10 (Krizhevsky, 2009).
3. Tuning a CNN on SVHN (Netzer et al., 2011) with varying number of layers, batch size, and number of filters (see Appendix A.4 for more details).

In the case of the SVM task, the allocated resource is number of training datapoints, while for the CNN tasks, the allocated resource is the number of training iterations.

We note that Fabolas was specifically designed for data points as the resource, and hence, is not directly applicable to tasks (2) and (3). However, freeze-thaw Bayesian optimization (Swersky et al., 2014), which was specifically designed for models that use iterations as the resource, is known to

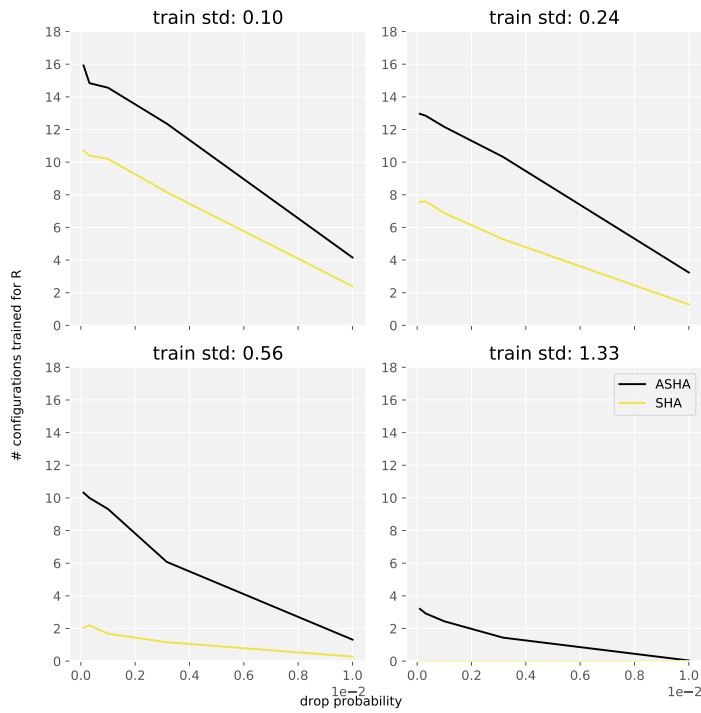

Figure 7: Average number of configurations trained on the maximum resource $R$ within 2000 time units for different standard deviations in training time and drop probabilities.

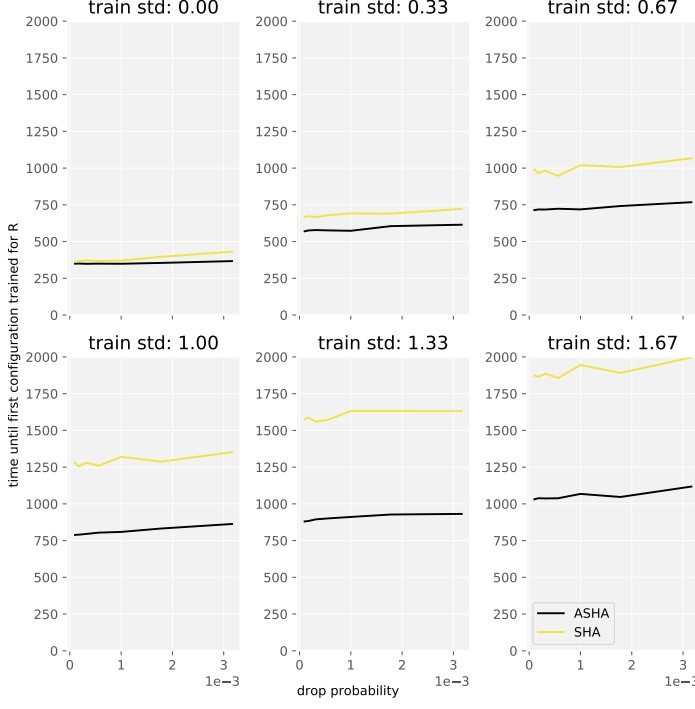

Figure 8: Average time before a configuration is trained on the maximum resource $R$ for different standard deviations in training time and drop probabilities.

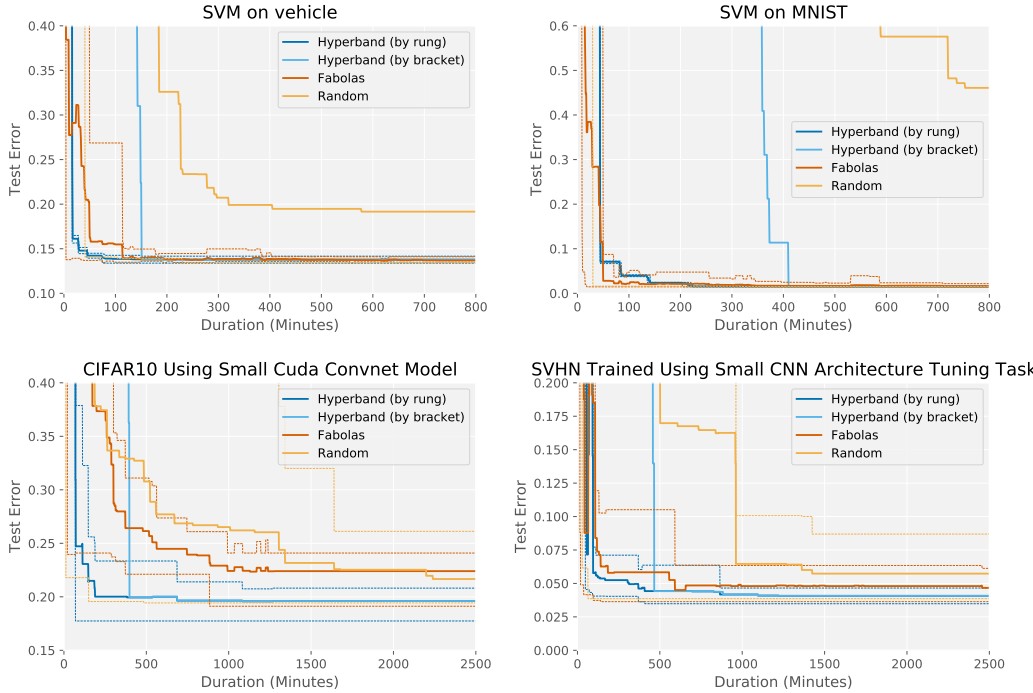

Figure 9: **Sequential Experiments** (1 worker) with Hyperband running synchronous SHA. Hyperband (by rung) records the incumbent after the completion of a SHA rung, while Hyperband (by bracket) records the incumbent after the completion of an entire SHA bracket. The average test error across 10 trials of each hyperparameter optimization method is shown in each plot. Dashed lines represent min and max ranges for each tuning method.

perform poorly on deep learning tasks (Domhan et al., 2015). Hence, we believe Fabolas to be a reasonable competitor for tasks (2) and (3) as well, despite the aforementioned shortcoming.

We use the same evaluation framework as Klein et al. (2017), where the best configuration, also known as the *incumbent*, is recorded through time and the test error is calculated in an offline validation step. Following Klein et al. (2017), the incumbent for Hyperband is taken to be the configuration with the lowest validation loss and the incumbent for Fabolas is the configuration with the lowest predicted validation loss on the full dataset. Moreover, for these experiments, we set $\eta = 4$ for Hyperband.

Notably, when tracking the best performing configuration for Hyperband, we consider two approaches. We first consider the approach proposed in Li et al. (2018) and used by Klein et al. (2017) in their evaluation of Hyperband. In this variant, which we refer to as "Hyperband (by bracket)," the incumbent is recorded *after the completion of each SHA bracket*. We also consider a second approach where we record the incumbent *after the completion of each rung* of SHA to make use of intermediate validation losses, similar to what we propose for ASHA (see discussion in Section 3.3 for details). We will refer to Hyperband using this accounting scheme as "Hyperband (by rung)." Interestingly, by leveraging these intermediate losses, we observe that Hyperband actually outperforms Fabolas.

In Figure 9, we show the performance of Hyperband, Fabolas, and random search. Our results show that Hyperband (by rung) is competitive with Fabolas at finding a good configuration and will often find a better configuration than Fabolas with less variance. Note that Hyperband loops through the brackets of SHA, ordered by decreasing early-stopping rate; the first bracket finishes when the test error for Hyperband (by bracket) drops. Hence, most of the progress made by Hyperband comes from the bracket with the most aggressive early-stopping rate, i.e. bracket 0.

## A.3 Experiments in Section 4.1 and Section 4.2

We use the usual train/validation/test splits for CIFAR-10, evaluate configurations on the validation set to inform algorithm decisions, and report test error. These experiments were conducted using `g2.2xlarge` instances on Amazon AWS.

For both benchmark tasks, we run SHA and BOHB with $n = 256$, $\eta = 4$, $s = 0$, and set $r = R/256$, where $R = 30000$ iterations of stochastic gradient descent. Hyperband loops through 5 brackets of SHA, moving from bracket $s = 0, r = R/256$ to bracket $s = 4, r = R$. We run ASHA and asynchronous Hyperband with the same settings as the synchronous versions. We run PBT with a population size of 25, which is between the recommended 20–40 (Jaderberg et al., 2017). Furthermore, to help PBT evolve from a good set of configurations, we randomly sample configurations until at least half of the population performs above random guessing.

We implement PBT with truncation selection for the exploit phase, where the bottom $20\%$ of configurations are replaced with a uniformly sampled configuration from the top $20\%$ (both weights and hyperparameters are copied over). Then, the inherited hyperparameters pass through an exploration phase where $3/4$ of the time they are either perturbed by a factor of 1.2 or 0.8 (discrete hyperparameters are perturbed to two adjacent choices), and $1/4$ of the time they are randomly resampled. Configurations are considered for exploitation/exploration every 1000 iterations, for a total of 30 rounds of adaptation. For the experiments in Section 4.2, to maintain 100% worker efficiently for PBT while enforcing that all configurations are trained for within 2000 iterations of each other, we spawn new populations of 25 whenever a job is not available from existing populations.

| Hyperparameter | Type | Values |
|---|---|---|
| batch size | choice | $\{2^6, 2^7, 2^8, 2^9\}$ |
| # of layers | choice | $\{2, 3, 4\}$ |
| # of filters | choice | $\{16, 32, 48, 64\}$ |
| weight init std 1 | continuous | $\log\left[10^{-4}, 10^{-1}\right]$ |
| weight init std 2 | continuous | $\log\left[10^{-3}, 1\right]$ |
| weight init std 3 | continuous | $\log\left[10^{-3}, 1\right]$ |
| $l_2$ penalty 1 | continuous | $\log\left[10^{-5}, 1\right]$ |
| $l_2$ penalty 2 | continuous | $\log\left[10^{-5}, 1\right]$ |
| $l_2$ penalty 3 | continuous | $\log\left[10^{-3}, 10^2\right]$ |
| learning rate | continuous | $\log\left[10^{-5}, 10^1\right]$ |

Table 1: Hyperparameters for small CNN architecture tuning task.

Vanilla PBT is not compatible with hyperparameters that change the architecture of the neural network, since inherited weights are no longer valid once those hyperparameters are perturbed. To adapt PBT for the architecture tuning task, we fix hyperparameters that affect the architecture in the explore stage. Additionally, we restrict configurations to be trained within 2000 iterations of each other so a fair comparison is made to select configurations to exploit. If we do not impose this restriction, PBT will be biased against configurations that take longer to train, since it will be comparing these configurations with those that have been trained for more iterations.

## A.4 Experimental Setup for the Small CNN Architecture Tuning Task

This benchmark tunes a multiple layer CNN network with the hyperparameters shown in Table 1. This search space was used for the small architecture task on SVHN (Section A.2) and CIFAR-10 (Section 4.2). The # of layers hyperparameter indicate the number of convolutional layers before two fully connected layers. The # of filters indicates the # of filters in the CNN layers with the last CNN layer having $2 \times$ # filters. Weights are initialized randomly from a Gaussian distribution with the indicated standard deviation. There are three sets of weight init and $l_2$ penalty hyperparameters; weight init 1 and $l_2$ penalty 1 apply to the convolutional layers, weight init 2 and $l_2$ penalty 2 to the first fully connected layer, and weight init 3 and $l_2$ penalty 3 to the last fully connected layer. Finally, the learning rate hyperparameter controls the initial learning rate for SGD. All models use a fixed learning rate schedule with the learning rate decreasing by a factor of 10 twice in equally spaced

| Hyperparameter | Type | Values |
|---|---|---|
| batch size | discrete | $[10, 80]$ |
| # of time steps | discrete | $[10, 80]$ |
| # of hidden nodes | discrete | $[200, 1500]$ |
| learning rate | continuous | $\log [0.01, 100.]$ |
| decay rate | continuous | $[0.01, 0.99]$ |
| decay epochs | discrete | $[1, 10]$ |
| clip gradients | continuous | $[1, 10]$ |
| dropout probability | continuous | $[0.1, 1.]$ |
| weight init range | continuous | $\log [0.001, 1]$ |

Table 2: Hyperparameters for PTB LSTM task.

intervals over the training window. This benchmark is run on the SVHN dataset (Netzer et al., 2011) following Sermanet et al. (2012) to create the train, validation, and test splits.

## A.5 Experimental Setup for Large-Scale Benchmarks

| Hyperparameter | Type | Values |
|---|---|---|
| learning rate | continuous | $\log [10, 100]$ |
| dropout (rnn) | continuous | $[0.15, 0.35]$ |
| dropout (input) | continuous | $[0.3, 0.5]$ |
| dropout (embedding) | continuous | $[0.05, 0.2]$ |
| dropout (output) | continuous | $[0.3, 0.5]$ |
| dropout (dropconnect) | continuous | $[0.4, 0.6]$ |
| weight decay | continuous | $\log [0.5e - 6, 2e - 6]$ |
| batch size | discrete | $[15, 20, 25]$ |
| time steps | discrete | $[65, 70, 75]$ |

Table 3: Hyperparameters for 16 GPU near state-of-the-art LSTM task.

The hyperparameters for the LSTM tuning task comparing ASHA to Vizier on the Penn Tree Bank (PTB) dataset presented in Section 4.3 is shown in Table 2. Note that all hyperparameters are tuned on a linear scale and sampled uniform over the specified range. The inputs to the LSTM layer are embeddings of the words in a sequence. The number of hidden nodes hyperparameter refers to the number of nodes in the LSTM. The learning rate is decayed by the decay rate after each interval of decay steps. Finally, the weight initialization range indicates the upper bound of the uniform distribution used to initialize all weights. The other hyperparameters have their standard interpretations for neural networks. The default training (929k words) and test (82k words) splits for PTB are used for training and evaluation (Marcus et al., 1993). We define resources as the number of training records, which translates into the number of training iterations after accounting for certain hyperparameters.

For the task tuning a modern LSTM architecture, we use the code provided by Merity et al. (2018) and construct a search space around the hyperparameter setting that they used. The hyperparameters that we considered along with their associated ranges are shown in Table 3.

