# OpenReview forum: "Massively Parallel Hyperparameter Tuning"
_ICLR.cc/2019/Conference_

### Official Review · AnonReviewer2 · 2018-10-31
**Massively parallel hyperparameter tuning**

**Rating:** 5
**Confidence:** 4

**Review:**



Authors describe a massively parallel implementation of the successive halving algorithm (SHA). Basically, the difference between SHA and its asynchronous version ASHA, is that the later can promote configurations to the next rung, without having to wait that a previous rung is completed.

I think this is a valuable contribution, although I am not sure if it has the level required by ICLR. The technical contribution of the paper is minor: it is a simple modification to an existing  methodology. However, authors perform an extensive evaluation and show that the implementation reaches SOTA performance.

Authors list 5 bullets as contributions of this paper. Whereas it is clear that the main contribution is the ASHA algorithm, the rest of contributions are in fact results that show the validity of the first contribution. I mean those contributions are the experimental evaluation to prove ASHA works.

Can authors provide more details on what a configuration is? I could think of as a snapshot of a model (with fixed architecture and hyperaramenters), but I do not think my understanding is totally correct: the x-axis in Figure 1 has 9 configurations which are explored throughout rungs, hence the same configuration is evaluated many times?

Authors claim that the method can find a good configuration in the time is required to train a network. How is this possible?, I guess is because the evaluation of each configuration is done in a small validation subset, this, however is not stated by the authors. Also, depending on the size of the validation set and implementation details, this is not necessarily an "amazing" result.

Why results from Section 4.1 are a contribution?, what is the impact of these results? Basically you compare existing methodologies (plus the proposed ASHA method).



State-of-the ART (abstract)
Authors mention the theoretical benefits of SHA, but they do not emphasize these in the paper, can they elaborate on this?

---

### Official Review · AnonReviewer1 · 2018-11-02
**well written paper but with only little novelty and missing baseline comparison**

**Rating:** 5
**Confidence:** 4

**Review:**

The paper describes a simple, asynchronous way to parallelize successive halving. In a nutshell, this method, dubbed ASHA, promotes a hyperparameter configuration to the next rung of successive halving when ever possible, instead of waiting that all configurations of the current rung have finished. ASHA can easily be combined with Hyperband which iteratively runs different brackets of successive halving. The empirical evaluation shows that the proposed method outperforms in the most cases other parallel methods such as Vizier and population based training.


Overall, the paper is well written and addresses an interesting and important research problem.
However, the paper contains only little novelty and proposes a fairly straight-forward way to parallelize successive halving.
Even though it shows better performance to other approaches in the non-sequential setting, its performance seems to decay if the search space becomes more difficult (e.g CNN benchmark 1 and 2 in Section 4.2), which might be due to its increasing number of mispromoted configurations.

Besides that the following points need to be clarified:

1) I am bit worried that asynchronous Hyperband performs consistently worse than asynchronous successive halving (ASHA). Why do we need Hyperband at the first place if a fixed bracket for ASHA seems to work better?
   It would be interesting to see, how the performance of ASHA changes if we alter its input parameters.

2) Why is PBT not included for the large-scale experiments in Section 4.3 given that it is a fairly good baseline for the more difficult search space described in Section 4.2?

3) Even though I follow the authors' argument that the parallel version of Hyperband described in the original paper is slower than ASHA / asynchronous Hyperband, it doesn't promoted suboptimal configurations and hence might achieve a better performance at the end. Why is it not included as a baseline?

4) I am also missing a reference and comparison to the work by Falkner et al., that introduced a different way to parallelize Hyperband which can make use of all workers at any time.


Falkner, Stefan and Klein, Aaron and Hutter, Frank
BOHB: Robust and Efficient Hyperparameter Optimization at Scale
Proceedings of the 35th International Conference on Machine Learning (ICML 2018)

---

### Official Review · AnonReviewer3 · 2018-11-02
**Good paper, but I have queries about the experiments**

**Rating:** 6
**Confidence:** 4

**Review:**

In this paper, the authors propose an extension of the Successive Halving Algorithm (SHA) allowing its deployment in a distributed setting, a so-called massively parallel setting. The proposed algorithm is relatively straightforward and the main contribution appears to be in the experimental validation of the method.

Quality: The work appears to be of a good quality. The experiments could benefit from some more rigorous statistical analysis, but that would most likely require a higher number of repetitions, which is understandably hard to do in a large-scale setting.

Clarity: In general, the paper is well written. However, the presentation of Algorithm 3.2 was confusing. More comments on this below.

Originality: The contribution is incremental, but important.

Justification: The problem is an important one. There is room for more research exploring the optimization of hyperparameters for large architectures such as deep networks.

Overall I think the paper is good enough for acceptation, but I found some elements that deserve attention. The experimental section is a bit perplexing, mostly in the first experiments on the sequential approaches. The final experiment on the large-scale setting is disappointing because it only compares ASHA with an underpowered version of Vizier, so the demonstration is not as impressive as it could be. Furthermore, PBT was discarded based on results from a small-scale benchmark, which were not very convincing either (possibly significantly better on one of two versions of the CIFAR-10 benchmark). If the authors have other reasons as to why PBT was not a good candidate for comparison, they should bring them forth.

Find below some more comments and suggestions:

Regarding Algorithm 3.2, promotability (line 12) is never defined explicitly, so this can lead to confusion (it did in my case). I think promotability also requires that at least eta jobs are finished in the rung k, am I correct? If so it is missing from the definition. Perhaps a subroutine should be defined?

It feels odd to me that the first experiment mentioned in the paper is tucked away in Appendix 1. I find it breaks the flow of the paper. The fact that SHA outperforms Fabolas I believe is one of the important claims of the paper. Hence, the result should probably not be in an Appendix. I would suggest putting Figure 3 in the Appendix instead, or removing/condensing Figure 2, which is nice but wastes a lot of space. I also fail to grasp the difference between the CIFAR-10 benchmarks in Appendix 1 and those in Section 4.2. It seems they could be joined in one single experiment comprising SHA v Hyperband v Fabolas v PBT (perhaps removing a variant of Hyperband to reduce clutter).

I also do not think the comparison between SHA and ASHA in a sequential case is relevant. The behavior of ASHA in the distributed case will be different than in the sequential case, so the comparison between the two variants of SHA does not bring any useful information. If I followed the method correctly, in the 9 worker example with eta=3, the first round of jobs would be all jobs at the lowest bracket (s=0), which would be followed by a round of jobs at the next bracket (3 jobs at s=1), and so on. Hence, the scheduling behavior would be exactly the same as SHA (albeit distributed instead of sequential). Am I correct in my assessment? If so, perhaps ASHA should just be removed from the sequential experiments.

As a point of sale, it might be interesting to provide the performance of models with manually tuned hyperparameters as a reference (i.e., the recommended hyperparameters in the reference implementations of those works that were cited).

Appendix A.2 serves no purpose and should probably be removed.

Section 4.3 & 4.3.1: In both cases, what is the running time R for a single model?

---

### Author Response · Authors · 2018-11-20
**Author Response (Part 3)**

Other Comments:
- (AnonReviewer3) “As a point of sale, it might be interesting to provide the performance of models with manually tuned hyperparameters as a reference (i.e., the recommended hyperparameters in the reference implementations of those works that were cited).”
In response to this comment, we’ve added a comparison for the large-scale LSTM experiment in Section 4.3, noting that the best model found by ASHA achieved a test perplexity of 76.6 compared to the published result of 78.4 for the large model in Zaremba et.al., 2015.  The comparison to previously published values is already provided in Section 4.3.1.

- (AnonReviewer1) “I am bit worried that asynchronous Hyperband performs consistently worse than asynchronous successive halving (ASHA). Why do we need Hyperband at the first place if a fixed bracket for ASHA seems to work better?”
In the JMLR Hyperband paper, the empirical results on CNNs and kernel classification also showed that the speedups for Hyperband, with the exception of the LeNet example in Section 3.3, came primarily from the most aggressive bracket.  We further confirm this in our experiments and view aggressive downsampling necessary for high dimensional search spaces, where we need to evaluate a multitude of configurations in order to adequately explore the search space.  As stated in the introduction, our goal is to evaluate orders of magnitude more hyperparameter configurations than available parallel workers.  Hence we focus on SHA with aggressive downsampling and present results for Hyperband (async) for completeness only.

- (AnonReviewer3) “Regarding Algorithm 3.2, promotability (line 12) is never defined explicitly, so this can lead to confusion (it did in my case). I think promotability also requires that at least eta jobs are finished in the rung k, am I correct? If so it is missing from the definition. Perhaps a subroutine should be defined?”
We have added a subroutine to clarify the definition of promotable configurations.

- (AnonReviewer1) “I am also missing a reference and comparison to the work by Falkner et al., that introduced a different way to parallelize Hyperband which can make use of all workers at any time.”
We have added this reference.  We’ve also added a comparison to BOHB as well as parallel version of SHA used in Falkner et. al. to the benchmarks in Section 4.2.   We also note that our work is contemporaneous with BOHB and in fact they cite our ICLR 2017 submission.

- (AnonReviewer2) “Can authors provide more details on what a configuration is? I could think of as a snapshot of a model (with fixed architecture and hyperaramenters), but I do not think my understanding is totally correct: the x-axis in Figure 1 has 9 configurations which are explored throughout rungs, hence the same configuration is evaluated many times?”
A configuration is simply a hyperparameter setting.  SHA/ASHA requires there to be a concept of partial training, either on subsets of the data or for a certain number of iterations.  A configuration can be visited multiple times by the algorithm if it is promoted to higher rungs and trained with a larger resource.  In the case of iterative algorithms with incremental training, the state of the model can be checkpointed after each rung so that if a configuration is promoted to the next rung, the training can be resumed from where it left off.

- (AnonReviewer2) “Authors claim that the method can find a good configuration in the time is required to train a network. How is this possible?”
Note that in each rung, a configuration is trained with a specific resource and evaluated on a validation set.  The total training resource that can be allocated to a configuration is R and with enough workers, ASHA will be able to return a configuration trained on R in time(R) with incremental training and slightly more without incremental training.

- (AnonReviewer2) “Authors mention the theoretical benefits of SHA, but they do not emphasize these in the paper, can they elaborate on this?”
Li et.al., 2018 showed that SHA requires a small multiple resource more than the optimal strategy in order to identify the best configuration (the optimal strategy would allocate just enough resource to each configuration to distinguish it from the best); this amount is often orders-of-magnitude less than that required by random search.  As discussed in Section 3.3, we expect ASHA to benefit from the same theoretical guarantees as SHA since the number of incorrect promotions diminishes as the rungs widen.

We have also addressed minor comments in the revised version of our paper.

---

### Author Response · Authors · 2018-11-20
**Author Response (Part 2)**

Parallel Experiments:
- (AnonReviewer1) “Why is PBT not included for the large-scale experiments in Section 4.3 given that it is a fairly good baseline for the more difficult search space described in Section 4.2?”
- (AnonReviewer3) “PBT was discarded based on results from a small-scale benchmark, which were not very convincing either (possibly significantly better on one of two versions of the CIFAR-10 benchmark). If the authors have other reasons as to why PBT was not a good candidate for comparison, they should bring them forth. “
At the time of running the experiments in Section 4.3, PBT had just been released and certain constraints prevented us from adding PBT subsequently.  That said, in response to these comments we have performed a comparison of PTB to ASHA for the modern LSTM tuning task with DropConnect in Section 4.3.1.  For this benchmark, ASHA finds a much better final result (lower is better); the best configuration trained to completion (i.e., 256 epochs) by ASHA achieves a validation perplexity of 60.6 and a test perplexity of 58.4 compared to 62.7 (validation) and 60.3 (test) for PBT.  We will add more trials for ASHA and PBT before the rebuttal deadline.

- (AnonReviewer1) “Even though I follow the authors' argument that the parallel version of Hyperband described in the original paper is slower than ASHA / asynchronous Hyperband, it doesn't promoted suboptimal configurations and hence might achieve a better performance at the end. Why is it not included as a baseline?”
For the experiments in Section 4.2, in response to this comment, we have added parallel synchronous SHA (both with and without Bayesian Optimization (BO)), where the work for each rung is distributed and new brackets are added when no jobs are available for existing brackets (this is the parallelization scheme used in the BOHB paper).  Overall, ASHA outperforms both of these methods. ASHA  finds a good configuration twice as fast on benchmark 1 (though BOHB finds a slightly better final configuration). Moreover, ASHA significantly outperforms synchronous SHA and BOHB on benchmark 2 due to the higher variance in training times between configurations, which exacerbates the sensitivity of synchronous SHA to stragglers. Note that BOHB actually underperforms synchronous SHA on benchmark 2.

We note that these experiments are conducted in very friendly environments with no dropped jobs and little hardware variance.  Hence, we compare ASHA and synchronous SHA on simulated workloads in Appendix A.1 and show ASHA evaluates many more configurations to completion and returns a configuration trained to completion faster than synchronous SHA when there are stragglers and dropped jobs.

We will add a comparison to BOHB for the sequential experiments in Section 4.1 before the rebuttal deadline.

- (AnonReviewer3) “The final experiment on the large-scale setting is disappointing because it only compares ASHA with an underpowered version of Vizier, so the demonstration is not as impressive as it could be.”
For the large-scale experiment on PTB, while we were only able to compare to Vizier without early-stopping, our results demonstrate the fragility of relying on a performance model for early-stopping.  Indeed, we attempted to alleviate the Vizier issue by capping perplexity to 1000 but Vizier with early-stopping still underperformed.

- (AnonReviewer1) “Even though it shows better performance to other approaches in the non-sequential setting, its performance seems to decay if the search space becomes more difficult (e.g CNN benchmark 1 and 2 in Section 4.2), which might be due to its increasing number of mispromoted configurations.”
We disagree with the remark that the performance of ASHA decays in the parallel setting.  Our results in Section 4.2 show that the average accuracy reached by ASHA with 25 workers in the parallel setting is the same as that in the sequential setting on the CudaConvnet benchmark and exceeds the result in the sequential setting on the small architecture tuning task.  Additionally, as demonstrated in the small architecture tuning task, ASHA achieves linear scaling with the number of workers if the tuning problem is difficult enough and warrants exploring number of configurations on the order of (# workers * \eta^(\log_\eta(R/r))).

- (AnonReviewer3) Section 4.3 & 4.3.1: In both cases, what is the running time R for a single model?
Due to certain proprietary restrictions for the large-scale benchmark in Section 4.3, we can only show timing in terms of the average time to train a configuration on R resource.  For the experiment in Section 4.3.1, it takes 20 hours to train a single model for the maximum resource of 256 epochs.

---

### Author Response · Authors · 2018-11-20
**Author Response (Part 1)**

We thank the reviewers for their feedback and comments.  Please see below for our responses and clarifications for some of the points and questions that were brought up.

Novelty:
- (AnonReviewer2) “I think this is a valuable contribution, although I am not sure if it has the level required by ICLR. The technical contribution of the paper is minor: it is a simple modification to an existing  methodology.”
- (AnonReviewer1) “However, the paper contains only little novelty and proposes a fairly straight-forward way to parallelize successive halving.”
Although our method is simple, we believe it is a novel approach to SHA and, as discussed in Section 3.3, is a significant improvement over the original algorithm in practice.  Moreover, we view the simplicity of our method to be a major asset, making it easy to implement and understand.  The novelty of ASHA is also apparent in that other papers that use parallel Hyperband, do not parallelize in an asynchronous way; existing approaches (i.e., BOHB) parallelize SHA by rung, leaving the algorithm highly susceptible to stragglers and dropped jobs (we have added Appendix A.1 to demonstrate this).  Additionally, these approaches run individual brackets of SHA, meaning the estimate of the top 1/eta configurations in each rung does not improve as more brackets are run. We also note that our work is contemporaneous with BOHB and in fact they cite our ICLR 2017 submission.

Sequential Experiments:
- (AnonReviewer3) “I also do not think the comparison between SHA and ASHA in a sequential case is relevant. The behavior of ASHA in the distributed case will be different than in the sequential case, so the comparison between the two variants of SHA does not bring any useful information.”
- (AnonReviewer2) “[W]hat is the impact of these results [in Section 4.1]?”
Section 4.1 serves to justify parallelizing SHA since it is competitive with SOTA hyperparameter optimization methods in the sequential setting (i.e. Fabolas and PBT).  Additionally, we compare ASHA with SHA in the sequential setting to demonstrate that premature promotions do not hurt ASHA.  While the promotions for ASHA with a single machine are certainly different than that with multiple machines, there will still be premature promotions.  In fact, ASHA with one worker will promote configurations earlier than ASHA with more workers.  In your 9 worker example, ASHA would make a promotion after seeing 9 configurations in the bottom rung, but ASHA with 1 worker would make a promotion after seeing 3 configurations in the bottom rung.  With more workers, ASHA will have more observations per rung before making a promotion decision but will be somewhat biased against configurations that take longer to complete initially; this bias will be corrected once longer running configurations complete.  Hence, we believe these sequential experiments are helpful in capturing the effect of early promotions although the effect of stragglers will not be present.

- (AnonReviewer3) “It feels odd to me that the first experiment mentioned in the paper is tucked away in Appendix 1. I find it breaks the flow of the paper. The fact that SHA outperforms Fabolas I believe is one of the important claims of the paper. Hence, the result should probably not be in an Appendix.”
We present the comparison with Fabolas in the appendix due to the specific evaluation scheme used by Fabolas.  Since Fabolas almost never trains configurations on the maximum resource, they track the current incumbent as determined by Fabolas and present results for the incumbent evaluated on the full resource in an offline evaluation step.  For consistency, we keep that evaluation scheme in the experiments in Appendix A.1.  For all the experiments shown in the main text of the paper, we use the usual evaluation scheme of reporting the best performance of a model actually evaluated by the hyperparameter optimization routine.  We agree that it breaks the flow to discuss the comparison to Fabolas first and have moved it to the end of Section 4.1.

---

### Author Response · Authors · 2018-11-26
**Revision Uploaded**

We’ve updated the paper with the additional experiments promised in our response.

Additionally, we’d like to re-emphasize the following points from our response:
1. Although ASHA is simple, we believe it is a novel approach to SHA and, as discussed in Section 3.3, is a significant improvement over the original algorithm in practice.
2. We demonstrate in Section 4.2 that ASHA is better suited for the parallel setting than parallelizing SHA by rung as is done by Falkner et.al., 2018.
3. We have added PBT to our modern LSTM benchmark in Section 4.3.1.

---

> ### Comment · AnonReviewer1 · 2018-11-29
> **Post rebuttal**
>
> Thanks for putting extra work into the rebuttal and adding the missing comparisons to existing baselines. I will increase my initial score from 4 to 5, but due to the little novelty, I do not think that the paper is yet above the threshold of being accepted at ICLR.

---

### Meta-Review · Area_Chair1 · 2018-12-10
**Solid experiments, nice and simple method, but too incremental**

**Confidence:** 3
**Recommendation:** Reject

**Metareview:**

The paper proposes and evaluates an asynchronous hyperparameter optimization algorithm.

Strengths:  The experiments are certainly thorough, and I appreciated the discussion of the algorithm and side-experiments demonstrating its operation in different settings.  Overall the paper is pretty clear.  It's a good thing when a proposed method is a simple variant of an existing method.

Weaknesses:  The first page could have been half the length, and it's not clear why we should care about the stated goal of this work.  Isn't the real goal just to get good test performance in a small amount of time?  The title is also a bit obnoxious and land-grabby - it could have been used for almost any of the comparison methods.  The proposed method is a minor change to SHA.  The proposed change is kind of obvious, and the resulting method does have a number of hyper-hyperparameters.

Consensus:  Ultimately I agree with the reviewers that is just below the bar of acceptance.  This does seem like a valid contribution to the hyperparameter-tuning literature, but more of an engineering contribution than a research contribution.  It's also getting a little bit away from the subject of machine learning, and might be more appropriate for say, SysML.